# Work hard and sleep better: Work autonomy attenuates the longitudinal effect of workaholism on sleep problem among Chinese working adults

workaholism; sleep problem; work autonomy; Chinese workers; evening work

**Corresponding author:**
Anise M.S. Wu;
Email: anisewu@um.edu.mo

Meng Xuan Zhang[1], Anise M.S. Wu[2,3] [ORCID], Ling Z. Zhang[4] and Long W. Lam[3,4]

[1]Department of Medical Humanities, School of Humanities, Southeast University, Nanjing, China; [2]Department of Psychology, Faculty of Social Sciences, University of Macau, Macao, China; [3]Centre for Cognitive and Brain Sciences, University of Macau, Macao, China and [4]Department of Management and Marketing, Faculty of Business Administration, University of Macau, Macao, China

## Abstract

The prevalence of workaholism has negative consequences on human health. Lack of sleep, a well-known problem among adults in modern society, is often attributed to overwork as a result of workaholism. Yet there is a lack of empirical research examining how and when workaholism will lead to sleep problems. To answer this question and to examine the longitudinal effect of workaholism on sleep in China, we investigate the mediating role of perceived evening responsibilities of work and the moderating effect of work autonomy. Two hundred and five Chinese working adults (58.0% female) voluntarily completed the online questionnaires at Time 1 (T1) and Time 2 (T2; 1-month later). Results showed that workaholism at T1 had a significant and positive correlation with sleep problem at T2. Further analysis suggested that perceived evening responsibilities of work fully mediated the relationship between workaholism and sleep problem. Work autonomy was shown to buffer the positive effect of workaholism on perceived evening responsibilities of work and attenuate the indirect effect of workaholism on sleep problem. While workers should be made aware of the negative impact of workaholism on sleep, organizations should also consider interventions to enhance employees' autonomy and control of their work.

## Impact statement

Workaholism is a common phenomenon among modern workers that can harm your health and productivity. Sleep problem, another phenomenon affecting millions of workers worldwide, is posited to be related to workaholism. This study reveals whether and how workaholism can affect your sleep quality, an important aspect of your well-being, over time. We surveyed Chinese working adults twice and found that workaholism led to sleep problems by increasing the perceived evening responsibilities of work. In other words, workaholics feel they have to work even after leaving the office, which prevents them from relaxing and sleeping well. On the other hand, we also found that work autonomy can buffer the negative effects of workaholism by giving workers more control and flexibility over their work. Workers with more autonomy can decide when, where and how they work, reducing their stress and improving their sleep quality. This study contributes to the literature that workaholism influences sleep via work-related perceptions. Modifying these perceptions (e.g., work autonomy) as well as some organizational policies (e.g., work hours and night work) thus may reduce the adverse impacts of workaholism on workers' health and well-being. This study's findings provide practical implications for workers, managers and policymakers who are concerned about the well-being of the workforce in a rapidly changing and competitive environment.

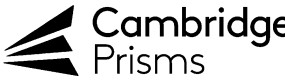



## Introduction

The term workaholism originates from Oates (1971), which has been referred to as "…the compulsion or the uncontrollable need to work incessantly" (p. 11). Research on workaholism has suggested that it comprises two factors: working excessively and working compulsively, with the former referring to spending more time on work-related activities beyond the need for organizational and economic requirements, and the latter emphasizes the obsession to work and a sense of loss of control over it even when they not working (Schaufeli et al., 2008). The latter is considered to be the addictive feature of workaholism, being driven by the workers' inner needs and compulsions rather than external rewards (Shimazu et al., 2010). As such, workaholism has

also been referred to as work addiction, including symptoms similar to other addictive behaviors. Specifically, workaholism as an addiction includes symptoms such as loss of control over when to do work and consistent preoccupation with work. The addictive symptoms and related outcomes of workaholism have thus given rise to growing public health concerns, including work–family conflicts (Tahir and Aziz, 2019), reduced job satisfaction (Caesens et al., 2014), deteriorated health (Schou Andreassen et al., 2007) and poor sleep quality (Kubota et al., 2014).

Workaholism is a rather prevalent phenomenon among working adults. A review of workaholism studies suggested that the percentage of workaholic employees can be as high as one-third of the working population (Sussman, 2012). More employees were reported as workaholics recently. A study in Korea revealed that the prevalence of workaholism was 39.7% in a sample of 4,242 adult employees (Kang, 2020). To address the public concerns about workaholism, the purpose of our study is to investigate the influence of workaholism on sleep and its underlying mechanism among Chinese working adults. Specifically, we conducted a longitudinal study by measuring sleep problem 1 month after measuring workaholism, and examined the hypothesized moderated mediation model (see Figure 1). Our investigation is important for several reasons. First, while we are aware of the adverse impact of workaholism on sleep, little is known about how workaholism could lead to such problem. Our investigation of perceived evening responsibility as the mediator may shed light on this issue. Second, overtime in China has been increasingly common and employees were reported to contribute about 10 hours above their normal work hours per week (National Bureau of Statistics of China, 2021). Surprisingly, while the extent of workaholism in China is believed to be higher than that in European countries (Hu et al., 2014), there is virtually no research examining employees' workaholism in China. Third, studies have shown that Chinese adults suffer from poor sleep quality which is another public concern (e.g., Li et al., 2018; Ma et al., 2022). By examining the buffering role of work autonomy (e.g., Ten Brummelhuis et al., 2017), our study can inform researchers and organizations how to accentuate the negative influence of workaholism on sleep among Chinese employees.

### Workaholism and sleep problem

Sleep problem has been regarded as a common symptom of many medical diseases and disorders, including cancer (Wong and Fielding, 2011; Matthews and Wang, 2022) and depression (Nutt et al., 2008).

Its prevalence in the adult population has been rising and high before and after the COVID-19 pandemic (e.g., 27% in young adults [Becker et al., 2018] and 35% in the general population [Dragioti et al., 2022]). Considering such high prevalence and its linkages with other health problems, scholars have thus considered sleep problem to be one of the most challenging health issues causing enormous burdens in modern society (Bhaskar et al., 2016; Becker et al., 2018). While sleep problem has less to do with complex sleep disorders such as sleep apnea and night terrors, it usually refers to the difficulty of falling asleep and/or maintaining sleep (Buysse et al., 1989). Previous empirical research has shown a significant association between workaholism and sleep problem such as insomnia, short sleep duration and poor sleep quality (e.g., Andreassen et al., 2018). A 7-month longitudinal study in Japan revealed that workaholism was associated with higher levels of sleep latency and daytime dysfunction (Kubota et al., 2014). Moreover, since workaholics tend to be obsessed with their work and cannot stop thinking about work, such thoughts may still linger during their bedtime and interfere with their capacity to fall to sleep (Van den Broeck et al., 2011; Salanova et al., 2016). These reasons lead us to expect a positive association between workaholism and sleep problem among Chinese employees.

### The roles of work responsibilities and work autonomy

For the relationship between workaholism and sleep problem, work responsibility in the evening is expected to play a vital role. Specifically, we expect workaholic employees to perceive more workload after completion of their normal work schedule (e.g., nine-to-five schedule) and feel the need to complete work during the evenings and even the nighttime (Kaiser et al., 2019). We further expect individuals with high perceived work responsibilities in the evening to have trouble falling asleep and maintaining sleep. That is because a key distinguishing feature of workaholism is work compulsively (Shimazu et al., 2010). If workaholics perceive a high workload in the evening, they may continue to work at the expense of time for sleep. This would contribute to more sleep problems. Moreover, Leroy's (2009) research suggested that people have attention residue from the previous task which affects how they perform the subsequent task. The attention residue effect is shown to be stronger when the previous task is not finished. Thus, when workaholic employees work in the evening, the attention residue left from work is likely to interfere with their sleep at night. A previous study provided support to this attention residue effect by showing that employees doing shift work would report more sleep disturbance due to their evening/

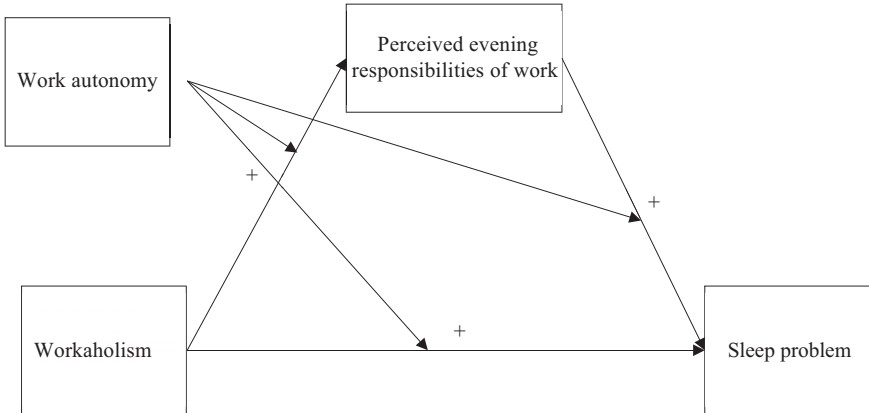

**Figure 1.** The hypothesized model.

night work schedule (Kerkhof, 2018). These reasons lead us to expect that workaholic employees, because of their perceived evening responsibility, will suffer from sleep problems.

We further posit that work autonomy, refers to perceived freedom and control over task arrangement (Spector, 1986), to attenuate the hypothesized relationship between workaholism and sleep via perceived evening responsibilities. According to the Job Demand-Control Model (Karasek, 1979), job control can buffer the impact of job demands on the strain and improve job satisfaction and well-being (Gameiro et al., 2020). Based on this model, researchers found employees with higher work autonomy to have better work productivity and job performance and report high job satisfaction and commitment (Khoshnaw and Alavi, 2020). We thus expect the proposed relationship (workaholism is positively related to sleep problem through perceived evening responsibilities (i.e., workaholism → perceived evening responsibilities → sleep problem) to be weaker at high work autonomy but stronger at low work autonomy.

Specifically, with high work autonomy, individuals have more freedom and control over the pace of their work (i.e., when and how to do it). While workaholic employees are still compelled to do their work, with high work autonomy, they may be more able to modify their work schedules in order to work more efficiently (Ramaswamy and Rajeesh Viswanathan, 2020). Thus, they may perceive less need to engage in work every evening and night, and as such, are less likely to suffer from sleep problems. On the other hand, when workaholic employees perceive low autonomy for their work, they may experience less control over their work schedules. Thus, their compulsion to work may still persist in the evenings, which may interfere with their sleep cycles (Tucker et al., 2015). As perceived evening responsibilities contribute to more sleep problems, we expect the mediated or indirect effect of workaholism on sleep problem to be less (more) pronounced under high (low) work autonomy.

## The hypotheses of the present study

This longitudinal study aims to empirically test the adverse influence of workaholism on sleep and a potential moderated mediation effects by evening responsibilities and work autonomy on such influence among Chinese working adults. Based on Figure 1, below is a summary of our hypotheses:

**Hypothesis 1 (H1):** There is a positive association between workaholism and sleep problem.
**H2:** Perceived evening responsibilities is positively associated with sleep problem.
**H3:** Perceived evening responsibilities mediate the relationship between workaholism and sleep problem.
**H4:** Work autonomy moderates the mediated relationship between workaholism and sleep problem via perceived evening responsibilities so that the mediated relationship is stronger at low rather than high levels of work autonomy.

## Methods

### Participants and procedures

Employees with at least 6 months of work experience were recruited from So-jump, a crowdsourcing platform for surveying working adults in China. In the first survey (T1), participants voluntarily answered the anonymous questionnaire, including workaholism, work autonomy, perceived evening responsibilities and demographical variables). In the second survey (T2), respondents were asked to report their sleep problem. The company (So-jump) made initial contact with 330 working adults and 205 of them (58.0% female, 85.9% aged 21–40 years old) completed surveys of both T1 and T2.

Before filling out the questionnaires in both waves, all participants were recruited by voluntary principle and provided their consent forms before the formal study. Their rights in the research (e.g., the confidentiality of their private information and dropping out of this research at any time without punishment) have been clarified in advance. The study procedures were carried out in accordance with the Declaration of Helsinki. The Institutional Review Board of the Department of Psychology at the corresponding author's affiliated university approved the study.

### Measures

All the scales used in this study were translated into Chinese version following Brislin's (1970) suggestion. During the process of translation and back-translation process by the bilingual research assistants, we also asked the opinions of working adults to construct the items in Chinese and conducted a pilot study pre-testing the translated measures.

### Workaholism

Following previous research (Taris et al., 2005), we used the Compulsive Tendencies subscale of Work Addiction Risk Scale (Robinson, 1999) to measure workaholism. This scale comprised eight items with a sample item "I put myself under pressure with self-imposed deadlines when I work." Participants rated all items on the 7-point scale (1 = "Strongly Disagree" to 7 = "Strongly Agree"). A higher total score represents a higher level of workaholism. The Cronbach's alpha of the scale was .78 in this study.

### Perceived evening responsibilities of work

We measure perceived evening responsibilities of work based on two items: "A good amount of my work took place during evenings or nights" and "I found myself to be the busiest at work during the evenings or nights" on a 7-point scale (1 = "Strongly Disagree" to 7 = "Strongly Agree"). The scale was developed based on Härmä et al.'s (2017) and Cully's (1998) studies on work time and responsibility. A higher total score indicates a higher level of perceived evening responsibilities of work and the scale had a good reliability ($\alpha$ = .85) in this study.

### Work autonomy

Similar to previous research (Spretizer, 1995), we used Hackman and Oldham's (1975) scale to measure work autonomy. The scale has three items (e.g., "I can decide on my own how to go about doing my work") capturing the degree of individual's choice and discretion involved in a job. Participants rated on 7-point scale from "Strongly Disagree" to "Strongly Agree." The Cronbach's alpha of the scale was .78 in this study.

### Sleep problem

Following Ten Brummelhuis et al. (2017), we measured the participants' sleep problem based on a scale developed by Van Veldhoven and Meijman (1994). Participants rated items such as "I wake up several times during the night" on a 7-point Likert scale from 1 (Strongly Disagree) to 7 (Strongly Agree). The internal reliability was .87 in this study.

### Demographics

Gender (1 = male, 2 = female), age (1 = ≤ 20 years to 8 = >51 years), educational level (1 = Primary school education to 6 = Graduate or above) and tenure (1 = half year or less to 6 = 10 years or above) were also reported in the surveys.

### Statistical analyses

In this study, the data of those participants who completed both baseline and follow-up surveys were included in the data analyses. Consistent with previous empirical studies (e.g., Bhandari et al., 2017; Zhao et al., 2022), the proposed moderated mediation model (Figure 1) was first examined with PROCESS in SPSS 26, with 5,000 replications in bootstrapping (Hayes, 2009). The effects of gender, age, education level and tenure on all the variables were controlled for during the model testing. Either the *p*-value or 95% confidence intervals (CI) were reported for examining the significance of the results.

### Results

#### Univariate correlations with sleep problem

For demographic effects, educational level was negatively associated with sleep problem ($r = -.21$, $p < .001$), while age was significantly and positively correlated with sleep problem ($r = .15$, $p < .05$). Gender was not significantly correlated with any variables except perceived evening responsibilities ($r = -.15$, $p < .05$).

Consistent with our expectation, workaholism (T1) showed a significant and positive correlation with sleep problem (T2) ($r = .23$, $p < .001$). It also significantly and positively associated with perceived evening responsibilities (T2) ($r = .44$, $p < .001$), which had a significant and positive correlation with sleep problem (T2) ($r = .39$, $p < .001$). in the correlation matrix is displayed in Table 1.

#### The moderated mediation model

Controlled for all the demographic effects, our analysis shows that workaholism was significantly related to perceived evening responsibilities (T2; $\beta = .44$, 95% CI = (.34, .54)). After controlling for the effect of workaholism, perceived evening responsibilities were also significantly related to sleep problem ($\beta = .34$, 95% CI = (.18, .50)). We thus had evidence supporting H1 and H2.

While we did not find a significant direct effect of workaholism (T1) on sleep problem (T2) ($\beta = .07$, 95% CI = (−.07, .21)), our results showed that the relationship between workaholism (T1) and sleep problem (T2) was mediated by perceived evening responsibilities (T1), with workaholism's indirect effect ($\beta$) = .15 and 95% CI = (.08, .24). Thus, H3 was supported.

Our results also showed work autonomy (T1) having a significant moderating effect on the relationship between workaholism (T1) and perceived evening responsibilities (T1) ($\beta = -.02$, 95% CI = (−.04, −.01)). For the other two paths (perceived evening responsibilities on sleep problem; workaholism on sleep problem), work autonomy did not have any significant moderating effect. With this result, we plotted the relationship between workaholism and perceived evening responsibilities at the high and low levels of work autonomy (see Figure 2).

Consistent with H4, the positive association between workaholism (T1) and perceived evening responsibilities (T1) was weaker among the participants with a higher level of work autonomy (T1) ($\beta = .08$, 95% CI = (.02, .14)) than those with the lower level of work autonomy (T1) ($\beta = .22$, 95% CI = (.16, .28)). Based on the final moderated mediation model (Figure 3), the conditional indirect effect of workaholism (T1) on sleep problem (T2) via perceived evening responsibilities (T1) was computed. The effect was weaker at high level ($\beta = .06$, 95% CI = (.03, .16)) but stronger at low level of work autonomy ($\beta = .17$, 95% CI = (.07, .28)). This evidence was supportive of H4.

### Discussion

To our best knowledge, this study is one of the pioneering research to test the longitudinal effect of workaholism on sleep problem among Chinese working adults. We also investigated the mediating effect of perceived evening responsibilities and the buffering effect of work autonomy. As hypothesized, workaholism had a significant and positive association with sleep problem via perceived evening responsibilities. This finding is consistent with prior studies on workaholism (Nam and Lee, 2019; Dutheil et al., 2020). We observed no gender difference in workaholism and sleep problem in our Chinese worker sample, suggesting that both male and female workers are equally prone to difficulties falling to sleep and maintaining sleeping quality if they are workaholics. Considering that workaholism is driven by inner obsession, workaholics

**Table 1.** The descriptive information and correlations of all variables (*N* = 205)

| | Mean | SD | 1 | 2 | 3 | 4 | 5 | 6 | 7 | 8 |
|---|---|---|---|---|---|---|---|---|---|---|
| 1. Sleep problem (T2) | 19.16 | 8.24 | – | | | | | | | |
| 2. Workaholism (T1) | 31.94 | 8.18 | .23*** | – | | | | | | |
| 3. ER (T1) | 8.48 | 4.52 | .39*** | .44*** | – | | | | | |
| 4. Work autonomy (T1) | 15.73 | 3.31 | −.25*** | .12 | −.07 | – | | | | |
| 5. Educational level | 4.92 | 0.61 | −.21** | −.06 | −.23*** | −.01 | – | | | |
| 6. Tenure | 4.82 | 1.10 | .11 | .01 | −.05 | .07 | −.26*** | – | | |
| 7. Age[a] | 3.94 | 1.46 | .15* | .03 | −.04 | .01 | −.15* | .78*** | – | |
| 8. Gender[b] | – | – | −.03 | .03 | −.15* | −.09 | .10 | .01 | −.06 | – |

ER, perceived evening responsibilities of work; SD, standard deviation; T1, Time 1; T2, Time 2.
*$p < .05$,
**$p < .01$, and
***$p < .001$.
[a]1 = ≤20 years (0.5%), 2 = 21–25 years (13.2%), 3 = 26–30 years (32.2%), 4 = 31–35 years (23.9%), 5 = 36–40 years (16.1%), 6 = 41–45 years (7.3%), 7 = 46–50 years (4.4%), 8= > 50 years (2.4%).
[b]1 = Male and 2 = Female.

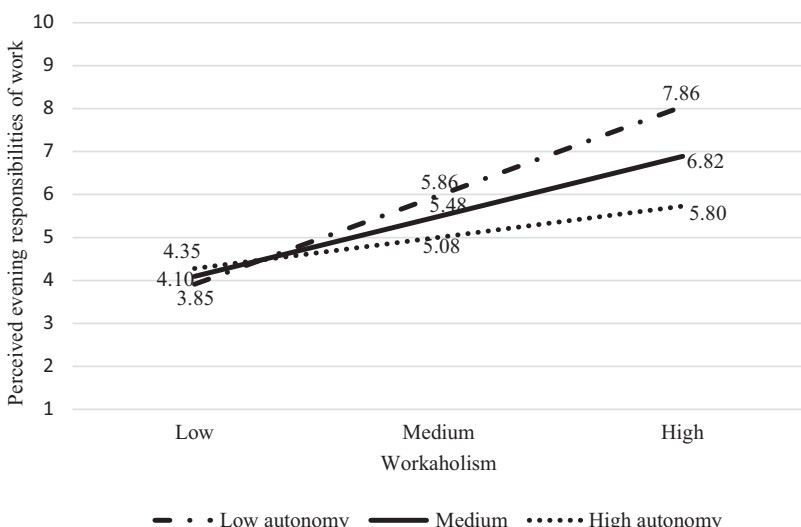

**Figure 2.** The moderating effect of autonomy between workaholic and perceived evening responsibilities of work.

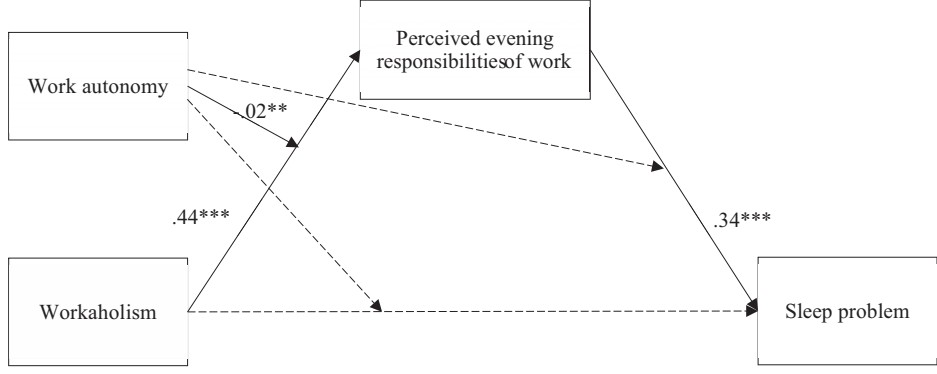

**Figure 3.** The moderated mediation model.
*Note:* **p < .01, ***p < .001.

should realize their tendency of workaholism and its negative impact on sleep health and seek help from professional counseling.

For the underlying mechanisms of the relationship between workaholism and sleep problem, this study found that perceived evening responsibilities of work served as the mediator of such relationship. Workaholism was positively associated with perceived evening responsibilities of work, which was associated with more sleep problems among Chinese workers. Since workaholics are obsessed with their work compulsively (Huml et al., 2021), their sense of work responsibilities will linger after work, and are likely to prioritize their work over other activities (Song and Lee, 2021). If they decide to work in the evenings and at nights, they will risk disturbing the biological rhythms by changing their sleep patterns (Song and Lee, 2021). Disturbance of sleep patterns will then give rise to sleep problem, including long sleep latency, short sleep duration and/or poor sleep quality.

Our results showed that work autonomy can attenuate the relationship between workaholism and perceived evening responsibilities of work so that workaholism has a less adverse impact on sleep. Proponents of the Job-Demand-Control model contend that job control in the form of work autonomy can buffer the negative impact of job demands on strain (Gameiro et al., 2020) and reduce the negative influence of workaholic emotions on emotional exhaustion (Spagnoli and Molinaro, 2020). Consistent with the

model's expectation, this study found that work autonomy could also buffer the effect of workaholism on perceived evening responsibilities of work. Workers, even if they have more obsession to work than others, will have more control over their work time to better align their work hours with their nonwork arrangements when they have higher work autonomy (Tucker et al., 2015). Thus, we believe that work autonomy may help workers reduce work–sleep interference.

Both workaholism and perceived evening responsibilities are risk factors for sleep quality. Behavioral change interventions may help workers to reestablish a healthy lifestyle (Weinstein et al., 2019) and reduce work addiction. Relevant organizational regulations may also be set to prevent excessing working in the evenings or at night. Employers should be aware of the negative impact of workaholism and workload in the evenings on sleep and work-health balance. As work autonomy can buffer the relationship between workaholism and perceived evening responsibilities of work, managerial interventions are needed to enhance the perceived control of work in order to allow workers to have more autonomy to schedule their work, and the ways to accomplish them. Although we emphasized the importance of work autonomy, other factors in the Chinese culture may also play a role in the workaholism–sleep relationship. For example, people in a collectivistic culture value connectedness with significant others (Potipiroon and Faerman, 2020). Support from family and

friends may prevent workaholic employees from suffering from resource strain or sleep problems. Thus, researchers may further test whether social support can serve as a moderator in the workaholism–sleep relationship.

This study has several limitations. First, our convenience sample with most of the participants in the younger generation may limit the generalizability of the findings to the entire workforce. Some participants also only participated in the first but not the second round of the survey. More worker samples in China should be collected in order to evaluate the effect of workaholism in future research. Second, the measurement of variables (e.g., workaholism and sleep problem) relies on self-report questionnaires, and most of the scales were originally developed and validated in Western societies, which may be susceptible to response biases and subjective judgment. These scales could be further validated among the Chinese working population. Further studies can also consider the use of objective measures (e.g., sleep applications in smartphone devices; see Grigsby-Toussaint et al., 2017) in addition to self-reported data. Third, we collected survey data from the general working population and did not have access to clinical samples with sleep or addictive disorders. Since we did not obtain data of workaholism and sleep problem via the clinical diagnosis of participants, future studies are encouraged to adopt the clinical perspective in assessing and evaluating these variables. Fourth, the one-month interval may not be long enough to fully reveal the impact of workaholism on sleep. To understand the full impact of workaholism, researchers may consider studies based on longer time periods and other techniques of data analysis (e.g., cross-lagged panel model) to further test the workaholism–sleep problem relationship as well as its autoregressive effects.

Despite these limitations, this study is the first to show the negative longitudinal effect of workaholism on sleep health among Chinese working adults. Our study's findings reveal that careful work design in terms of work autonomy can buffer the positive effect of workaholism on perceived evening responsibilities. Organizations should consider enhancing work autonomy and work flexibility to order to promote work-health balance among workers.

## Conclusions

Workaholic employees are obsessed with work, which seems like a good thing to companies striving for productivity and profitability. But organizations should also take into account that some of the consequences of workaholism have to do with negative impacts on employees' well-being. For example, sleep quality is related to employees' physical and psychological conditions the next day (Schilpzand et al., 2018). Therefore, we are eager to know whether workaholic employees have significant sleep problem and how to buffer it. Our research finally reveals that workaholism is positively correlated with sleep problem. Further, we highlight that perceived evening responsibilities are the mediation mechanism between workaholism and sleep problem. However, work autonomy could buffer the positive effect of workaholism on perceived evening responsibilities of work and attenuate the indirect effect of workaholism on sleep problem. These hypotheses gained support from a field study. We hope our research stimulates future work to consider the research on the health-related consequences of workaholism and the associated buffering factors.

**Open peer review.** To view the open peer review materials for this article, please visit http://doi.org/10.1017/gmh.2023.68.

**Data availability statement.** Data are available on request due to privacy/ ethical restrictions.

**Author contribution.** M.X.Z. was involved in research conception, data analysis, literature review and manuscript writing. A.M.S.W. was involved in research conception, manuscript preparation and revision. L.Z.Z. was involved in questionnaire preparation and data collection. L.W.L. was responsible for the research conception and design, project coordination and manuscript revision. All authors contributed to and approved the final manuscript.

**Financial support.** This research received no specific grant from any funding agency, commercial or not-for-profit sectors.

**Competing interest.** The authors declare none.

**Ethics statement.** The ethics approval was obtained from the research ethics committee of the Department of Psychology at the University of Macau (DPSY2021-20).

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
