## [Reviewer Report]

Dear Editor, 

We are writing to submit a manuscript entitled “Work hard and sleep better: Work autonomy attenuates the longitudinal effect of workaholism on sleep problem among Chinese working adults” to the Global Mental Health for publication as an original article.

This manuscript reports the findings of an anonymous questionnaire survey that aimed to test the longitudinal effect of workaholism on sleep and investigate the mediating role of perceived evening responsibilities of work and the moderating effect of work autonomy among Chinese workers. The results showed that workaholism at T1 was significantly and positively correlated with sleep problem at T2. The bootstrapping analysis suggested that perceived evening responsibilities of work fully mediated the relationship between workaholism and sleep problem. Moreover, work autonomy could buffer the positive effect of workaholism on perceived evening responsibilities of work and attenuate the indirect effect of workaholism on sleep problem. Considering the adverse effect of workaholism on sleep, organizations should also consider interventions to enhance employees’ autonomy and control for their work. The research obtained ethics approval from the research ethics committee of the Department of Psychology, University of Macau.

This manuscript has not been published elsewhere and is not under any concurrent journal review. Should you need any further information, please contact Dr. Meng Xuan Zhang, Department of Medical Humanities, School of Humanities, Southeast University, Nanjing, 211189, Jiangsu, China. E-mail: zhangmengxuan@seu.edu.cn. Phone: +(86) 15122560962

Yours Sincerely, 

Prof. Anise M. S. Wu

Department of Psychology, Faculty of Social Sciences, University of Macau, Avenida da Universidade, Taipa, Macau, China. 

E-mail: anisewu@um.edu.mo

Phone: +(853) 8822 8377

Fax: +853-8822-2337

---

## [Reviewer Report]

This is a very interesting paper focusing on workaholism and sleep. Please consider revising it based on the comments below.

1. introduction

The researcher has noted in a previous study that The term workaholism originates from Oates (1971), which has been referred to as “...the compulsion or the uncontrollable need to work incessantly” (p.11). It is stated as follows. On the other hand, the current study design refers to changes in workaholism after one month. Considering the definition, can workaholism tendency change in such a short period of time?

P7.Methods

The number of subjects was 205. Is it possible to verify by post-test how many samples were originally needed? Please mention whether it was sufficient or not.

3. Is there any reason for limiting the sample to workers up to the age of 40? If so, please add a statement to that effect. If content, please provide a supplementary explanation in the limitations of the study. Are there any possible effects of targeting a relatively young generation?

P9. Methods

Is there any rationale for the 5000 for replication by the bootstrapping method? Please supplement any previous studies.

P12. limitation

Please mention any other possible confounding factors that may limit the study.

---

## [Reviewer Report]

Thank you very much for submitting this article. It is a very interesting work, novel for mental health interventions. My comments are in relation to:

General

1. It is important to consider the temporality of the study, since in recent years the impact of global situations such as the covid-19 pandemic also affected mental health and labor and economic regimes worldwide. This can be considered an important context with a greater or lesser effect depending on the time of the study.

Although studies conducted in Asian countries are cited, the scales used were mostly constructed and validated in Western countries. It is important to consider the differences in thinking and perception of the Chinese population and to mention other studies related to welfare and work culture. This is also relevant in the construction of the discussion.

Participants and procedures

3. What happened to the 125 participants not contacted at T2, what were the reasons for non-contact.

4. During T2, were they assessed again with the workaholism scale? If not, in what way is it determined that workaholism problems still persisted and thus could still have an effect on sleep? It is possible that other cultural factors may be playing a role in their sleep?

6. It is also important to know if the participants suffered from any physical or mental health comorbidity, or if they were already receiving some type of treatment for health care and addiction regulation.

Measures

7. How were the scales used adapted to the language? Were they answered in English or translated into Chinese? Was there professional intervention for the correct adaptation of each scale? Consider that this could affect the understanding of the questions and the scale.

Why did you use only a subscale of Robinson’s Work Addiction Risk Test for the measurement and not other short scales such as the Berger Work Addiction Scale? Which has recently been published a validation article in Chinese social workers.

9. Has the scale of perceived work responsibilities been constructed by the study, and if it already existed, cite its authors.

---

## [Reviewer Report]

I think “acceptable” is acceptable because the paper is well responded to the peer review opinion.